# Endoscopic Features of Chronic Rhinosinusitis in Patients with Gastroesophageal Reflux Disease

**DOI:** 10.3390/medicina60081257

**Published:** 2024-08-02

**Authors:** Kalamkas Sagandykova, Nataliya Papulova, Talapbek Azhenov, Aliya Darbekova, Bayan Aigozhina, Jerome R. Lechien

**Affiliations:** 1“University Medical Center” Corporate Fund, School of Medicine, Nazarbayev University (NUSOM), Astana 010000, Kazakhstan; 2Department of Otorhinolaryngology, NpJSC “Astana Medical University”, Astana 010000, Kazakhstan; papulova.n@amu.kz (N.P.); azhenov.t@amu.kz (T.A.); aigozhina.b@amu.kz (B.A.); 3Medical Center Hospital of the President’s Affairs Administration of the Republic of Kazakhstan, Astana 010000, Kazakhstan; darbekova@bmc.mcudp.kz; 4Laryngopharyngeal Reflux Study Group, Young-Otolaryngologists of the International Federations of Oto-Rhino-Laryngological Societies (YO-IFOS), 13005 Paris, France; jerome.lechien@umons.ac.be; 5Department of Otolaryngology-Head and Neck Surgery, Foch Hospital, 78180 Paris, France; 6School of Medicine, University Paris Saclay, 91190 Paris, France

**Keywords:** endoscopic picture, laryngopharyngeal reflux, rhinosinusitis, nasopharyngeal reflux, the Reflux Symptom Index, the Reflux Symptom Score

## Abstract

Chronic rhinosinusitis (CRS) is a complex inflammatory condition affecting the nasal and paranasal sinus mucosa. Gastroesophageal reflux disease (GERD) has been implicated as a potential exacerbating factor in CRS, but the specific endoscopic features of nasopharyngeal pathology in this context remain poorly understood. *Background and Objectives*: Chronic rhinosinusitis is a multifactorial disease with various underlying etiologies, including inflammation, anatomical factors, and environmental triggers. While gastroesophageal reflux disease has been suggested as a potential contributor to chronic rhinosinusitis, the specific endoscopic features indicative of nasopharyngeal pathology in CRS patients with GERD symptoms have not been clearly elucidated. Our aim is to identify specific endoscopic features of nasopharyngeal pathology in patients with CRS associated with GERD symptoms and to propose a method for assessing the influence of gastroesophageal reflux disease on the mucosal layer of the nose and nasopharynx. *Materials and Methods*: We conducted a cross-sectional observational study involving 521 adult patients presenting with symptoms suggestive of CRS. From this cohort, 95 patients with the highest scores on the Reflux Symptom Index (RSI) and Reflux Symptom Score-12 (RSS-12) questionnaires were selected as the main group. Endoscopic examinations were performed to assess the nasal and nasopharyngeal mucosa. *Results*: Our study revealed significant alterations in the nasopharyngeal mucosa of patients with CRS associated with GERD symptoms. Increased vascularity of the nasopharyngeal mucosa was observed in 91 patients (95.7%), while hypertrophy was noted in 83 patients (87.4%). Mucus was present in the nasopharynx of 77 patients (81.1%), exhibiting varying characteristics of color and consistency. Asymmetric hypertrophy of the oropharyngeal mucosa was noted in 62 patients (65.3%). *Conclusions*: We propose a method for assessing the influence of gastroesophageal reflux disease on the mucosal layer of the nose and nasopharynx, which may aid in diagnostic and management decisions. Further research is warranted to explore the potential impact of GERD symptoms on the course and severity of CRS exacerbations.

## 1. Introduction

Adult chronic rhinosinusitis (CRS) is an inflammation of the mucosal layer of the nose and paranasal sinuses, characterized by two or more symptoms, one of which must be either nasal congestion or nasal discharge (anterior or posterior). Additional symptoms may include facial pain and/or pressure and a decreased or lost sense of smell lasting for 12 weeks or more. The diagnostic criteria include endoscopic signs such as nasal polyps, mucopurulent discharge predominantly from the middle nasal passage, swelling/obstruction of the mucosa predominantly of the middle nasal passage, and/or computed tomography (CT) changes such as mucosal changes in the osteomeatal complex and/or sinuses [1]. Chronic rhinosinusitis significantly impacts the quality of life and healthcare systems, with an estimated prevalence of 10–12% in the general population [1].

Classification According to European Position Paper on Rhinosinusitis and Nasal Polyps (EPOS): Endotypes and Phenotypes. According to the EPOS, chronic rhinosinusitis with nasal polyps (CRSwNP) can be further classified into different endotypes based on underlying pathophysiological mechanisms, which include type 2 inflammation (eosinophilic) and non-type 2 inflammation (neutrophilic). The type 2 endotype is characterized by eosinophilic inflammation and elevated levels of interleukins IL-4, IL-5, and IL-13, whereas non-type 2 endotypes are less well defined and may involve neutrophilic inflammation. The pathophysiology of CRSwNP involves a complex interplay of genetic, environmental, and immunological factors leading to chronic inflammation. Epidemiological studies indicate that chronic rhinosinusitis with nasal polyps affects about 2–4% of the general population.

Numerous factors can cause and exacerbate chronic rhinosinusitis, making it refractory to optimized treatment. These factors include genotypic or phenotypic mucosal changes, scarring and synechiae, allergies, smoking, and gastroesophageal acid reflux [1]. The current definition of actionable gastroesophageal reflux disease (GERD) requires convincing evidence of reflux-related pathology, demonstrated by endoscopy and/or abnormal reflux monitoring using the Lyon Consensus thresholds, along with compatible bothersome symptoms. While typical bothersome symptoms alone may justify antisecretory drug trials, esophageal testing is recommended for all other symptom categories and for patients who do not respond to proton pump inhibitors (PPIs). This testing should be conducted prior to invasive gastroesophageal reflux disease treatment or long-term medical therapy [2]. Changes in the laryngeal mucosal layer due to laryngopharyngeal reflux were first described and systematized in 2001 by Belafsky et al. [3]. In 2020, Lechien et al. further detailed the endoscopic appearance of the larynx under the influence of laryngopharyngeal reflux by describing changes on the pharyngeal side [4]. In 2022, Zeleník et al. identified a relationship between hypertrophy of the inferior turbinate and extraesophageal reflux [5].

Treatment for chronic rhinosinusitis with nasal polyps typically includes a combination of medical and surgical approaches. Initial management often involves intranasal corticosteroids to reduce inflammation and polyp size [1,6]. Endoscopic sinus surgery (ESS) is considered for patients who do not respond to medical therapy, aiming to restore sinus ventilation and drainage [1,7].

Novel Therapeutics for chronic rhinosinusitis with nasal polyps. Recent advances in the understanding of the immunopathology of CRSwNP have led to the development of novel biologic therapies targeting specific inflammatory pathways. Biologics such as dupilumab (anti-IL-4Rα), mepolizumab (anti-IL-5), and omalizumab (anti-IgE) have shown promise in clinical trials, offering new hope for patients with refractory chronic rhinosinusitis with nasal polyps [8,9]. These therapies work by modulating the immune response, thereby reducing the polyp burden and improving symptoms and quality of life.

Endoscopic findings of laryngopharyngeal reflux have been well documented, but the specific endoscopic features of nasopharyngeal pathology in CRS patients with GERD symptoms remain poorly understood. Previous studies have highlighted the association between extraesophageal reflux and nasal mucosal hypertrophy, underscoring the need for further investigation into the relationship between gastroesophageal reflux disease and chronic rhinosinusitis.

The aim of our study is to comprehensively assess the endoscopic features of the nose and nasopharynx in patients with chronic rhinosinusitis who also exhibit symptoms of gastroesophageal acid reflux disease. Our specific objective is to detect any aberrations in the endoscopic presentation of chronic rhinosinusitis associated with symptoms of acid gastroesophageal reflux disease. We expect this study to contribute to improved care for patients with chronic rhinosinusitis by facilitating more accurate diagnostic and therapeutic strategies.

We hypothesize that patients with chronic rhinosinusitis and concomitant gastroesophageal acid reflux disease will exhibit distinct endoscopic features compared with those without reflux symptoms, indicating a potential link between gastroesophageal reflux and the exacerbation or persistence of chronic rhinosinusitis.

## 2. Materials and Methods

The Local Ethical Commission of “Astana Medical University” NpJSC approved the study protocol (LCB NpJSC AMU #13). The database is available as 10.6084/m9.figshare.25594425. 

### 2.1. Study Design and Population

The research method is a cross-sectional observational study design. A total of 521 adult patients from September 2023 to February 2024 with chronic rhinosinusitis were screened and treated at the University Medical Center Corporative Fund (UMC CF) and Multidisciplinary Hospital #1 in Astana, Kazakhstan. The patients met the clinical definition of chronic rhinosinusitis, characterized by symptoms caused by an inflammatory process in the nasal and paranasal sinus mucosa. Clinical symptoms included nasal congestion, nasal discharge, and/or facial pain/pressure, with or without a decreased or lost sense of smell lasting more than 12 weeks. The diagnosis was confirmed through computed tomography and endoscopic imaging.

All patients were examined using endoscopic diagnostic methods during the remission period, outside of exacerbations of chronic rhinosinusitis, and before starting anti-reflux treatment for gastroesophageal reflux disease. All patients underwent a clinical evaluation by a gastroenterologist, which included esophagogastroduodenoscopy and pH measurement of the esophagus.

Standardized questionnaires were administered to all patients with confirmed chronic rhinosinusitis and GERD symptoms. To determine the regularity of changes in the nasal and nasopharyngeal mucosa, we selected patients with the highest scores on the Reflux Symptom Index (RSI) (>20) and Reflux Symptom Score-12 (RSS-12) (>130). 

Patients who smoked, had active seasonal allergies or asthma, had a history of laryngeal cancer, or were pregnant were excluded from the study, resulting in a final group of 95 patients. The study included patients with no history of smoking for the last 5 years or more. During the study, two patients with suspected malignant tumors of the larynx were referred for consultation to an oncologist and were excluded from the study. Active seasonal allergies or asthma were determined based on the medical history. The medical history included data from an allergist’s examination, which comprised spirometry and skin tests.

The control group 1 included 41 patients with chronic rhinosinusitis but without GERD symptoms. The control group 2 consisted of 10 patients who showed no signs of chronic rhinosinusitis or GERD symptoms.

### 2.2. CRS Characteristics

Total Cohort (91 + 41 patients):

The clinical symptoms were as follows: nasal congestion (94.69%), nasal discharge (anterior or posterior) (56.81%), facial pain/pressure (36.36%), decreased or lost sense of smell (46.21%), and general discomfort (18.93%).

Classification was as follows: type 2 (eosinophilic), 34.1% vs. non-type 2 (neutrophilic), 65.9%; CRSwNP, 44.7% vs. CRSsNP, 55.3%.

Comorbidities were as follows: chronic otitis media, 26.51%; allergic rhinitis, 25.75%; asthma, 19.69%; nonsteroidal anti-inflammatory drug-exacerbated respiratory disease (NERD), 12.87%; and sleep disturbances, including obstructive sleep apnea (OSA), 5.3%.

The treatment history was as follows: one or more operations on the sinuses, 75.51%; antibacterial treatment with a course of more than 10 days, more than three times a year, in the last 5 years, 58.33%. All patients had been treated repeatedly for more than 1 month with topical corticosteroids.

The CT and endoscopic findings in our study correlated well with each other, providing additional information that improved the diagnostic accuracy and facilitated treatment planning.

Mucosal Thickening:

The CT findings were as follows: CT scans revealed mucosal thickening of the sinuses, a common feature of CRS.

The endoscopic findings were as follows: endoscopy revealed swollen and inflamed mucosa consistent with areas of mucosal thickening seen on CT scans.

Nasal Polyps:

The CT findings were as follows: CT scans showed the presence and extent of polyps in the nasal cavity and sinuses in patients with chronic polypoid rhinosinusitis.

The endoscopic findings were as follows: Endoscopy provided direct visualization of nasal polyps, allowing assessment of their size, location, and extent. However, in five patients diagnosed with sinus polyps on CT scans, no polyps were identified on endoscopic examination.

Ostiomeatal Complex Obstruction:

The CT findings were as follows: CT scans revealed ostiomeatal complex obstruction in 47 patients.

The endoscopic findings were as follows: endoscopy confirmed the presence of obstruction in the ostiomeatal complex, showing thick pus or polypoid tissue obstructing the drainage pathways.

### 2.3. Standardized Questionnaires

The Reflux Symptom Index (RSI), a nine-item self-administered questionnaire developed by Belafsky et al. (2002) [3], was used to document the presence and degree of symptoms of laryngopharyngeal reflux. The maximum score on the RSI is 45 [10].

The Reflux Symptom Score (RSS-12) is a 12-item self-administered tool used to diagnose and monitor laryngopharyngeal reflux (LPR) and its impact on quality of life. The maximum score is 300 [11].

### 2.4. Endoscopic Assisment

A flexible endoscope was used to perform an endoscopy of the nasal cavity and nasopharynx. The condition of the lower nasal turbinate was assessed according to the Camacho classification [12,13], which graded the total airway space occupied by the turbinate as follows:Grade 1: 0–25%Grade 2: 26–50%Grade 3: 51–75%Grade 4: 76–100%

Additionally, the condition of the nasopharyngeal mucosa and the Eustachian tube junction was evaluated.

## 3. Results

### 3.1. Demographics

Table 1 presents the descriptive statistics of the groups under consideration in terms of the sex of the subjects. 

Table 2 presents the descriptive statistics for age, RSS-12, and RSI by group. In the main group, the mean age was 48.9 years, with a confidence interval (CI) of ±2.8 years. The youngest participants were in control group 2, with a mean age of 32.5 years and a CI of ±4.0 years.

For the RSS-12 score, the main group had a mean score of 29.0, with a CI of ±1.2, while control group 1 had a mean score of 9.8 with a CI of ±0.8, and control group 2 had a mean score of 7.6 with a CI of ±1.4.

Regarding the RSI score, the main group had a notably high mean value of 186.8, with a CI of ±7.4, whereas control group 1 and control group 2 had mean scores of 14.5 and 14.0, respectively, with CIs of ±1.4 and ±1.8.

Table 3 presents the results of comparing the means of age, RSS-12, and RSI between groups based on sex, race, and observation group. The *p*-value indicated whether the difference between the means was statistically significant. A *p*-value greater than 0.05 suggested that the means were not significantly different.

The Mann–Whitney U criterion for independent groups, noted as 1, was applied to determine significance. This non-parametric statistical criterion compared two independent samples on the level of a quantitatively measured characteristic. A smaller *p*-value suggested more reliable differences between the values of a parameter in the samples.

The Kruskal–Wallis criterion method for independent groups, noted as 2, was used to determine if there was a statistically significant difference between the medians of three or more independent groups, specifically the “main”, “control 1”, and “control 2” groups. This non-parametric test was chosen due to the violation of the assumption of normality of the data distribution in the groups. If the *p*-value was greater than 0.05, it indicated that there was no statistically significant difference between the medians.

### 3.2. Endoscopic Findings

During endoscopic examination of the nose and nasopharynx, we noted distinct alterations in the mucosa of the posterior parts of the nasal cavity, particularly at the posterior end of the inferior nasal concha, in patients with chronic rhinosinusitis associated with gastroesophageal reflux disease. Additionally, a notable contrast was observed in the condition of the nasal cavity mucosa between the anterior and posterior regions. In the anterior parts, the mucosa may exhibit no changes or demonstrate grade 1 hypertrophy based on the Camacho classification. Anterior dry rhinitis with crusts was frequently encountered.

Moving to the posterior regions of the nasal cavity, we observed severe edema, asymmetrical hypertrophy of the posterior ends of the lower nasal bones, and copious mucus production. Nasal edema was detected by two blinded raters in 75 patients (78.9%). This finding underscored the prevalence of nasal edema in our study population (see Figure 1A). However, discerning these findings as a specific characteristic of chronic rhinosinusitis associated with gastroesophageal disease in adults posed a challenge.

In comparison with the control groups, this symptom was prominent in the main group, manifesting as pronounced swelling of the nasal mucosa. Conversely, reactive nasal edema was not observed in control group 1, where hypertrophic changes without active nasal edema were noted during the remission period. Control group 2 showed no alterations in the nasal mucosa, as patients in this group did not exhibit chronic rhinosinusitis or gastroesophageal reflux disease.

Significant alterations were observed in the nasopharyngeal mucosal layer. Increased vascularity of the nasopharyngeal mucosal layer was noted in 91 patients (95.7%) (refer to Figure 1B,C). The underlying pathogenetic mechanism of this vascular pattern remains unknown. However, we hypothesized that it may be caused by thinning of the mucosal layer under the influence of reflux content.

Hypertrophy of the nasopharyngeal mucosal layer was observed in 83 patients (87.4%) (see Figure 1D). In 77 patients (81.1%), mucus with varying characteristics of color and consistency was found in the nasopharynx (see Figure 1E), while no mucus or other secretions were detected in the middle nasal passages in these patients. The color of the mucus ranged from transparent to a pronounced green hue. Its consistency was viscous, characterized by thick, difficult-to-remove mucosal discharge.

Asymmetric hypertrophy of the mucosa of the oropharynx was noted in 62 patients (65.3%) (see Figure 1F). The more pronounced lesion on one side of the nose and nasopharynx was associated with a preference for falling asleep and sleeping on either the right or the left side. Therefore, we propose that these hypertrophic changes in the nose and nasopharynx may be induced by the effect of acidic reflux content.

Based on the endoscopic findings described in the results, the criteria for diagnosing chronic rhinosinusitis associated with symptoms of gastroesophageal reflux disease include the following:

Alterations in nasal cavity mucosa:Posterior nasal cavity: Distinct alterations in the mucosa are present, particularly at the posterior end of the inferior nasal concha.Anterior nasal cavity: Mucosa may show no changes or grade 1 hypertrophy according to the Camacho classification. Anterior dry rhinitis with crusts may also be observed.Posterior nasal cavity: Severe edema, asymmetrical hypertrophy of the posterior ends of the lower nasal bones, and copious mucus production are present.

Nasal edema:Nasal edema detected by two blinded raters is present in a significant percentage of patients (78.9%).

Nasopharyngeal mucosal alterations:Increased vascularity: This is noted in a majority of patients (95.7%), indicating possible inflammation or irritation.Hypertrophy: This is observed in a high percentage of patients (87.4%), suggesting chronic inflammation.Mucus production: This is present in a majority of patients (81.1%), with varying characteristics of color and consistency.

Oropharyngeal asymmetry:Asymmetric hypertrophy of the mucosa of the oropharynx is noted in a significant percentage of patients (65.3%), potentially influenced by sleeping position preference.

These endoscopic findings, when observed in conjunction with symptoms suggestive of both chronic rhinosinusitis and gastroesophageal reflux disease, contribute to the diagnosis of chronic rhinosinusitis associated with symptoms of gastroesophageal reflux disease in adults.

After analyzing all the data collected, we concluded that in chronic rhinosinusitis associated with gastroesophageal disease, the mucosal layer of the posterior parts of the nose and nasopharynx underwent continuous inflammatory processes due to the regular influence of acidic reflux content.

## 4. Discussion

Our study findings indicated that gastroesophageal reflux exerted an influence on the mucosal layer of the nasal cavity and nasopharynx, akin to its effect on the larynx and pharynx, characterized by edema, mucus presence, and increased vascularization with hyperemia and hypertrophy. Several studies have underscored the etiopathogenetic role of gastroesophageal reflux in sinus and nasopharyngeal inflammation [14,15,16,17]. Analysis of the causal relationship between gastroesophageal reflux disease and chronic rhinosinusitis at the genetic level has revealed that gastroesophageal reflux disease increases the risk of developing chronic rhinosinusitis by 36% [18].

In our study, we identified specific abnormalities: significant changes in the mucosa of the posterior parts of the nose (including the posterior ends of the lower nasal turbinates and the nasopharyngeal region). Similar to findings in the pharynx [19], we also observed an increased vascular pattern and the presence of mucus in the posterior parts of the nose and nasopharynx. A study investigating the relationship between laryngopharyngeal reflux and otitis media with effusion in children demonstrated that pepsin levels gradually increased as the viscosity of the fluid in the middle ear cavity increased [15]. It is believed that exposure to gastric contents via nasopharyngeal reflux triggers hypersecretion of mucus in the nasopharynx.

The excessive production of mucus results in postnasal drip syndrome, which constitutes the primary source of discomfort for individuals with nasopharyngeal reflux, consequently diminishing their quality of life. This syndrome is typified by the drainage of nasal secretions from the nose, passing through the nasopharynx, and pooling at the posterior wall of the pharynx. Many patients often describe difficulty in clearing this viscous mucus when attempting to blow their nose or swallow.

In our study, we observed significant alterations in the nasopharyngeal and nasal mucosal layers. Various theories exist regarding how gastroesophageal reflux disease impacts the nasal and nasopharyngeal cavities. One hypothesis suggests that the acidic reflux contents may directly affect the nasal and nasopharyngeal mucosal layers, as evidenced by a reaction similar to that observed in the esophageal mucosa upon direct contact with gastric contents, including the expression of pepsin A and heat shock protein 70 [20]. Another theory implicates Helicobacter pylori in the development of chronic rhinosinusitis, potentially leading to the formation of nasal polyps [21]. Additionally, autonomic nervous system dysfunction associated with gastroesophageal reflux disease may contribute to the pathogenesis through an existing nerve reflex between the esophagus and the sinuses via the vagus nerve [21].

The characteristics of changes in the nasal and nasopharyngeal mucosa observed during endoscopic examination of the nose and nasopharynx in patients with nasopharyngeal reflux are presented in Table 4.

In addition to the above-mentioned results, we used the Camacho classification for the objective assessment of nasal cavity patency. In the second phase of our study, we plan to use the Lund–Kennedy endoscopic scoring system (LK) to objectively assess the severity of endoscopic manifestations in patients with chronic rhinosinusitis, both before treatment and after 8 weeks of therapy. The Lund–Kennedy scale rates polyps, discharge, edema, scarring, and crusting on a scale from 0 to 2 for each category, with higher scores indicating a more severe disease course. This scoring system is widely used in clinical practice and research to objectively assess the severity of chronic rhinosinusitis and track changes over time. It provides a standardized method to evaluate endoscopic findings, facilitating comparison across studies and improving the accuracy of the diagnosis and treatment outcomes [22].

Several studies have investigated the relationship between chronic rhinosinusitis and gastroesophageal reflux disease using the Lund–Kennedy endoscopic scoring system. For example, a study in 2022 by Yeo et al. examined laryngopharyngeal reflux symptoms and signs in CRS patients, utilizing the Lund–Kennedy scale for endoscopic assessment. The study showed that subjective LPR symptoms were associated with subjective CRS symptoms. However, the researchers noted that Lund–Mackay scores did not significantly correlate with the preoperative Reflux Symptom Index and Reflux Finding Score [23]. Another study conducted by DelGaudio et al. (2005) established a significant association between extraesophageal reflux and increased endoscopic scores in CRS patients, suggesting that reflux might exacerbate nasal and sinus inflammation [24].

These findings align with previous research, which suggests that reflux, particularly extraesophageal reflux, contributes to the pathophysiology of chronic rhinosinusitis, potentially worsening the endoscopic appearance of the disease. This evidence underscores the importance of assessing and managing reflux symptoms in patients with chronic rhinosinusitis to potentially improve overall treatment outcomes.

## 5. Conclusions

Our comprehensive study, encompassing 95 patients with chronic rhinosinusitis accompanied by symptoms of gastroesophageal reflux disease along with two meticulously chosen control groups, illuminated profound alterations in the mucosal landscape of the posterior nasal and nasopharyngeal cavities. These discerning findings not only shed light on the intricate interplay between chronic rhinosinusitis and gastroesophageal reflux disease but also furnished a practical framework for the nuanced differential diagnosis of the etiopathogenetic trajectory of chronic rhinosinusitis. By offering this pragmatic diagnostic approach, our research holds the potential to serve as a valuable tool for general practitioners and otolaryngologists, empowering them to precisely identify the underlying triggers of chronic rhinosinusitis. Through such accurate diagnoses, clinicians may be better equipped to implement tailored interventions, ultimately mitigating the frequency and severity of exacerbations in affected individuals. We posit that our study heralds a promising avenue for further exploration, advocating for continued research endeavors to deepen our understanding of this intricate relationship and optimize patient care strategies accordingly.

## Figures and Tables

**Figure 1 medicina-60-01257-f001:**
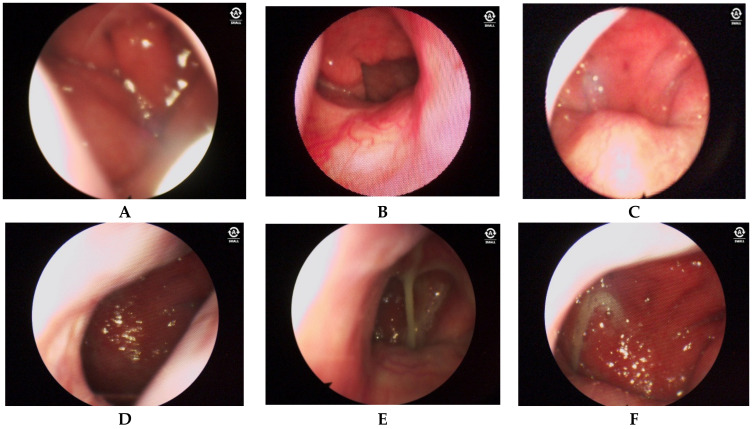
Endoscopic findings. A—Nasal edema. B,C—Increased vascularity. D—Hypertrophy of the nasopharyngeal mucosal layer. E—Mucus in the nasopharynx. F—Asymmetric hypertrophy of the mucosa of the oropharynx.

**Table 1 medicina-60-01257-t001:** Descriptive statistics of group structure by sex.

	Group		
	Main	Control 1	Control 2
Male (n = 75)	46	23	6
% in group	48.4	56.1	60
Female (n = 71)	49	18	4
% in group	51.6	43.9	40

**Table 2 medicina-60-01257-t002:** Descriptive data on age, RSS-12, and RSI by group.

	Main		Control 1		Control 2	
	Mean	CI	Mean	CI	Mean	CI
Age	48.9	2.8	36.8	3.2	32.5	4.0
RSS-12	29.0	1.2	9.8	0.8	7.6	1.4
RSI	186.8	7.4	14.5	1.4	14.0	1.8

**Table 3 medicina-60-01257-t003:** Descriptive data on age, RSS-12, and RSI by group.

		Age	*p*-Value ^1^	RSS-12	*p*-Value ^1^	RSI	*p*-Value ^1^
		Middle		Middle		Middle	
Sex	Male	43.23 (CI 3.29)	0.252	22.36 (CI 2.60)	0.641	125.39 (CI 21.47)	0.828
	Female	45.65 (CI 3.11)		21.87 (CI 2.34)		127.76 (CI 19.32)	
Race	M	44.75 (CI 2.77)	0.809	22.80 (CI 2.21)	0.075	128.07 (CI 18.14)	0.099
	E	43.49 (CI 3.89)		20.39 (CI 2.56)		122.63 (CI 22.15)	
Group	Main	48.94 (CI 2.77)	0.000 ^2^	28.96 (CI 1.20)	0.000 ^2^	186.76 (CI 7.40)	0.000 ^2^
	Control 1	36.78 (CI 3.19)		9.83 (CI 0.82)		14.46 (CI 1.42)	
	Control 2	32.50 (CI 3.99)		7.60 (CI 1.44)		14.00 (CI 1.85)	

^1^ Mann–Whitney U test for independent groups. ^2^ Kruskal–Wallis criterion for independent groups.

**Table 4 medicina-60-01257-t004:** Signs of exposure to nasopharyngeal reflux.

	Signs of Exposure to Nasopharyngeal Reflux	0	1	Total Score
Nose	Asymmetry between the anterior and posterior regions of the nasal cavity			
	Predominantly unilateral hypertrophy of the posterior end of the inferior turbinate Absence of mucus in the middle nasal passage			
Nasopharynx	Hypertrophy of the posterior wall of the nasopharynx			
	Hypertrophy of the Eustachian junction			
	Increased vascular pattern			
	Presence of mucus			

## Data Availability

Data is contained within the article.

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
