# Peer review of "Endoscopic Features of Chronic Rhinosinusitis in Patients with Gastroesophageal Reflux Disease"

_medicina, 2024, doi:10.3390/medicina60081257_

Round 1

Reviewer 1 Report

Comments and Suggestions for Authors

Dear authors, dear Editor,

The present study aims to explore the endoscopic features of the nasal cavity and nasopharynx in patients with chronic rhinosinusitis (CRS) who also exhibit symptoms of gastroesophageal reflux disease (GERD). The study is designed as a cross-sectional observational study, analyzing the endoscopic presentations in 95 patients with CRS and GERD compared to control groups. The study's main contributions include the identification of specific endoscopic abnormalities associated with GERD in CRS patients, which could enhance diagnostic precision and therapeutic strategies.

Major comments:

- The introduction of chronic rhinosinusitis is too weak to give readers a good understanding of the disease. Please expand the introduction including epidemiology, classification according to EPOS in endotypes and phenotypes, associated comorbidities like asthma or NERD, short pathophysiology, treatment options, novel therapeutics for CRSwNP including biologica.

- Change the first reference to Fokkens et al.European Position Paper on Rhinosinusitis and Nasal Polyps 2020. Rhinology. 2020 Feb 20;58(Suppl S29):1-464., and that CRS can be defined according to EPOS...

- please state the study hypothesis after the study's aim.

 - Results section:
1.) Change the subdheadings name from description into subdheading names explaining the content, like "demographics" or "endoscopic findings"

2.) Detailed clinical data of CRS charecteristics are missing. Please provide data for the total cohort and groups/subgroups including clinical symptoms, classification (type 2 vs Non-type 2 or CRSwNP vs CRSsNP, comorbidities, treatment history (oral cortiocsteroids, systemic corticosteroids, long-term antibiotics?), in CRSwNP cases: nasal polyp score, number of prior endoscopic sinus surgeries, known allergies, Lund-McKay Score in CT.
3.)Consider making sub groups Outcome analysis for CRSwNP and CRSsNP.
4.) Anny correlations between CT findings and Endoscopic findings? please provide.
5.) Have you also obtained QoL questionnaires regarding CRS? like the SNOT-22 ?

- Have you obtained the Lund-Kennedy endoscopic score? If not, please state this in the discussion section and discuss with previous studies.

- Line 242-246: This statement does not belong to the results section.

- Include a limitation paragraph

- Discussion is too short, expand and include more relevant literature.

Minor comments:

- Data availability statement ist missing

- Figure 1 legend: Please provde specific explanation of the panels A-D.

- Ensure that all abbreviations (e.g., CI, RSI, RSS-12) are defined when first used and that tables are referenced clearly in the text.

Author Response

Thank you for the detailed revision of our text. We have made every effort to incorporate all your corrections and recommendations. Below, I have attached our revised text based on your comments.

Reviewer 2 Report

Comments and Suggestions for Authors

Thank you for the opportunity to review this article. This is an interesting topic that affects a huge group of patients.

Unfortunately, the article needs improvement. Below are my comments.

1. The introduction and discussion need to be improved. At the moment, the introduction is of little interest and provides little, rather basic information. 

2. I think the statement
„We anticipate that this investigation will contribute to enhanced patient care by facilitating more precise diagnostic and therapeutic strategies, potentially mitigating the occurrence and duration of chronic rhinosinusitis exacerbations.” 
is misused, as the above study does not analyse treatment or the occurrence of exacerbations. 

3. As far as the bibliography of this work is concerned it is not very extensive, I think that the work would benefit if more works on the subject that are generally available were cited. 

4. The work is underdeveloped in terms of the abbreviations introduced, e.g. the abbreviation GERD is introduced more than once, when rhinosinusitis is protected for the first time no abbreviation is introduced and this abbreviation is often not used either, the abbreviation RSS is used once and RSS-12 once (both introduced separately), etc. - abbreviations need to be put in order. 

5. „All patients were examined using endoscopic diagnostic methods during the remission period of chronic rhinosinusitis and before starting anti-reflux treatment for gastroesophageal reflux disease (GERD).” 
- w
hat do you mean by "remission period"?

6. „GERD symptoms were confirmed in all patients 94 using endoscopic diagnostic methods”
- please clarify what the diagnostic methods were, how was the assessment carried out, was pH-metry performed?

7. Please define what you mean by 'Standardised questionnaires' - what information was included in this questionnaire. 

8. Active seasonal allergies or asthma were determined by history or spirometry, Skin prick tests?

9.  The term Control 1 and Control 2 used in Table 1 is not introduced in the text - I understand that this refers to patients without GERD symptoms and without GERD and CRS symptoms, but this is not clearly defined. 

10. The tables in the results can be combined, and the data that are in the tables do not need to be duplicated in the text - the text can be shortened.

11. Figure 2 needs to be described in more detail and the images should be merged into 1 image so that there are no A, B, C offsets - each small image needs to specify what it represents.

Author Response

(The authors gave the same response as above.)

Round 2

Reviewer 1 Report

Comments and Suggestions for Authors

The authors have sufficiently adressed my comments and the mansucript has improved.

Author Response

No comments

Reviewer 2 Report

Comments and Suggestions for Authors

The article is still underdeveloped, disorganised, abbreviations are introduced repeatedly, there are repetitions, punctuation and stylistic errors. Please read and correct the article carefully. Responses to my queries should also be included in the manuscript (5,6,8). The article needs corrections to be publishable. 

Author Response

Thank you for your attention. All comments have been addressed. Answers to questions 5, 6, and 8 have been included in the text of the article.